# Human iPSC-Derived 3D Hepatic Organoids in a Miniaturized Dynamic Culture System

**DOI:** 10.3390/biomedicines11082114

**Published:** 2023-07-26

**Authors:** Serena Calamaio, Marialaura Serzanti, Jennifer Boniotti, Annamaria Fra, Emirena Garrafa, Manuela Cominelli, Rosanna Verardi, Pietro Luigi Poliani, Silvia Dotti, Riccardo Villa, Giovanna Mazzoleni, Patrizia Dell’Era, Nathalie Steimberg

**Affiliations:** 1Cellular Fate Reprogramming Unit, Department of Molecular and Translational Medicine, University of Brescia, 25123 Brescia, Italy; serena.calamaio@grupposandonato.it (S.C.); marialaura.serzanti1@unibs.it (M.S.); 2Laboratory of Tissue Engineering, Department of Clinical and Experimental Sciences, University of Brescia, 25123 Brescia, Italy; jennifer.boniotti@unibs.it (J.B.); giovanna.mazzoleni@unibs.it (G.M.); 3Oncology and Experimental Immunology Unit, Department of Molecular and Translational Medicine, University of Brescia, 25123 Brescia, Italy; annamaria.fra@unibs.it; 4Laboratory Diagnostics, Department of Molecular and Translational Medicine, University of Brescia, 25123 Brescia, Italy; emirena.garrafa@unibs.it; 5Pathology Unit, Department of Molecular and Translational Medicine, University of Brescia, 25123 Brescia, Italy; manuelacominelli.83@gmail.com (M.C.); luigi.poliani@unibs.it (P.L.P.); 6Laboratory for Stem Cell Manipulation and Cryopreservation, Department of Transfusion Medicine, ASST Spedali Civili di Brescia, 25123 Brescia, Italy; verardirosanna@gmail.com; 7Istituto Zooprofilattico Sperimentale della Lombardia e dell’Emilia-Romagna, 25124 Brescia, Italy; silvia.dotti@izsler.it (S.D.); riccardo.villa@izsler.it (R.V.)

**Keywords:** liver, hiPSC, organoids, 3D dynamic culture, organotypic culture

## Abstract

The process of identifying and approving a new drug is a time-consuming and expensive procedure. One of the biggest issues to overcome is the risk of hepatotoxicity, which is one of the main reasons for drug withdrawal from the market. While animal models are the gold standard in preclinical drug testing, the translation of results into therapeutic intervention is often ambiguous due to interspecies differences in hepatic metabolism. The discovery of human induced pluripotent stem cells (hiPSCs) and their derivatives has opened new possibilities for drug testing. We used mesenchymal stem cells and hepatocytes both derived from hiPSCs, together with endothelial cells, to miniaturize the process of generating hepatic organoids. These organoids were then cultivated in vitro using both static and dynamic cultures. Additionally, we tested spheroids solely composed by induced hepatocytes. By miniaturizing the system, we demonstrated the possibility of maintaining the organoids, but not the spheroids, in culture for up to 1 week. This timeframe may be sufficient to carry out a hypothetical pharmacological test or screening. In conclusion, we propose that the hiPSC-derived liver organoid model could complement or, in the near future, replace the pharmacological and toxicological tests conducted on animals.

## 1. Introduction

Modern medicine has undergone a continuous and incessant growth in recent years, thanks to the great success achieved with laboratory research, which has introduced new tools in support of several disciplines, including regenerative medicine, pharmacology, and tissue engineering [1]. A major breakthrough occurred in 2007, when it was discovered that adult human cells could be rejuvenated to a pluripotent state. These embryonic-like cells had the potential to be further differentiated toward any cell type of the human body [2]. Fifteen years later, the study of cells derived from iPSCs, which are generated from healthy individuals or patients affected by genetic diseases, revealed the pathophysiology of many biochemical and molecular processes that are difficult to identify using animal models. Indeed, this model has become one the most valuable tools to fulfill the European “Three Rs” (3Rs) request to reduce, refine, and replace animal testing while overcoming the limitations associated with the use of human tissue [3].

The first protocol applied to differentiate hiPSCs was similar to the one employed to differentiate embryonic stem cells. It is based on the three-dimensional (3D) self-aggregation of cells, resulting in the formation of an embryoid body [4]. Further development of this concept led to the generation of the so-called “tissue-in-a-dish”, where specific differentiated cell types from a specific organ are mixed in a 3D structure that self-organizes, resembling the in vivo organogenesis process [5]. This advancement greatly increased the capacity of this model to be used as a tool for drug testing as well as for regenerative medicine [6].

Concerning the liver, hepatic cells derived from hiPSCs could represent an important source of cells to be used in preclinical tests as well as in cell-based therapeutic applications such as transplantation. In addition to difficulty in obtaining donor human hepatocytes, one of the major challenges is to preserve hepatocyte functions that are usually rapidly lost under extracorporeal conditions. Differentiation of hepatocyte-like cells from hiPSCs has already been performed in several laboratories [7,8]. However, the use of hiPSC-derived hepatocytes for assessing drug-induced hepatoxicity still needs to improve its predictability. Hepatic organoids represent a potent tool for sustaining hepatic functionality also because they better mimic the complexity of the liver both from an architectural point of view and as regards the multicellular and heterotypic composition of the organ [9,10].

Even though static culture conditions favor cell expansion, they do not facilitate the diffusion of nutrients in a 3D structure, particularly during long-term cultivation. To overcome this problem, the use of continuous fluid flow in new culture systems emerges as a viable solution. Bioreactor technology was developed to address this challenge by enabling dynamic perfusion, which ensures constant nutrient renewal and continuous removal of waste metabolites. This dynamic process closely mimics the exchange of physiological body fluids in vivo [11]. Bioreactors offer the ability to create a highly controlled environment, allowing the operator to adjust various cell culture conditions, such as pH, temperature, perfusion rate, and oxygen levels. Numerous studies have demonstrated that cultures maintained in bioreactors closely replicate the conditions found in living organs, thus providing the opportunity to conduct increasingly refined drug tests [9,12].

Here, we introduce a model that combines the heterotypic composition of liver cells (endothelial, mesenchymal, and hepatocyte cells) and the 3D organization of cells (organoids in 3D matrix), as well as the microfluidic simulation of sinusoidal flux within the intrahepatic microenvironment.

## 2. Materials and Methods

### 2.1. Cell Culture

All cultures were kept at 37 °C in a humidified atmosphere of 5% CO_2_.

#### 2.1.1. HUVECs

Human umbilical vein endothelial cells (HUVECs; Lonza, Basel, Switzerland) were used at early passages (I–IV) and grown on plastic Petri dishes coated with 0.1% porcine gelatin (Merck, Darmstadt, Germany) in Endothelial Cell Growth Medium-2 (EGM-2; Lonza) with 5% fetal bovine serum (FBS). Cells were passaged using trypsin–ethylenediaminetetraacetic acid (EDTA) (Thermo Fisher Scientific, Waltham, MA, USA) every 4 days.

#### 2.1.2. Human Induced Pluripotent Stem Cells (hiPSCs)

An hiPSC line from a healthy female donor (https://hpscreg.eu/cell-line/TMOi001-A, accessed on 21 June 2023) was bought from Thermo Fisher Scientific and used for all the experiments. Either mTeSR^TM^ or TeSR-E8^TM^ cell medium (STEMCELL Technologies, Vancouver, BC, Canada) was used for the maintenance of the cell line. Plates were coated with Human Biolaminin 521 LN (Thermo Fisher Scientific) for maintenance and Matrigel^®^ hESC-qualified Matrix (Corning, Corning, NY, USA) for differentiation. When cells reached 70–80% confluency they were detached using TrypLe^TM^ Select Enzyme (Thermo Fisher Scientific) and plated in the presence of Rock inhibitor Y27632 (STEMCELL Technologies) at a seeding density of around 3 × 10^4^ cells/cm^2^. The culture medium was refreshed daily.

#### 2.1.3. Human Induced Mesenchymal Stem Cells (hiMSCs)

Mesenchymal progenitors were derived from the hiPSC line from a healthy female donor (https://hpscreg.eu/cell-line/TMOi001-A, accessed on 21 June 2023). hiMSCs were maintained in MesenCult^TM^ ACF Plus Medium (STEMCELL Technologies). Cells were sub-cultured with trypsin–EDTA (Thermo Fisher Scientific) every 4 days. The culture medium was refreshed daily.

### 2.2. hiPSC Differentiation

#### 2.2.1. Hepatic Differentiation

The hepatic differentiation protocol sought to recapitulate the in vivo changes occurring during embryogenesis. hiPSCs were differentiated toward definitive endoderm (iDE) using a STEMdiff^TM^ Definitive Endoderm Kit (STEMCELL Technologies). For hepatic lineage specification, the medium was replaced with hepatocyte maturation medium consisting of RPMI 1640 + B27 Supplement + 10 ng/mL basic Fibroblast Growth Factor + 20 ng/mL Bone Morphogenetic Protein 4 (all from Thermo Fisher Scientific). From day 9 of differentiation (dd9), the human induced hepatocytes (hiHeps) were maintained in hepatocyte maintenance medium (HMM) consisting of Clonetics^®^ HCM^TM^ (Lonza) + 5% FBS + 100 nM Dexamethasone + 20 ng/mL Oncostatin M + 10 ng/mL Hepatocytes Growth Factor (all from STEMCELL technologies).

#### 2.2.2. Mesenchymal Differentiation and Characterization

Mesenchymal progenitors were derived from the hiPSC line using a STEMdiff^TM^ Mesenchymal Progenitor Kit (STEMCELL Technologies). hiPSCs were seeded on Matrigel^®^-coated 6-well multiwell plates two days before mesenchymal induction, in mTeSR^TM^ medium with daily change. After two days, the mTeSR^TM^ medium was replaced with STEMdiff^TM^ Mesenchymal Induction Medium to induce early mesoderm progenitors. After four days, the cells were subjected to a second induction process to derive early mesenchymal progenitor cells (MPCs), and the medium was replaced with MesenCult^TM^-ACF Plus Medium. After the second passage, the cells were seeded and maintained on plastic in MesenCult^TM^-ACF Plus Medium. hiMSCs were split twice a week with daily medium change. To verify MSC identity, several markers were tested in flow cytometry. Cells were further differentiated to adipocytes using StemPro^®^ Complete Adipogenesis Differentiation Medium (Thermo Fisher Scientific) or to osteoblasts using StemPro^®^ Complete Osteogenesis Differentiation Medium (Thermo Fisher Scientific). Adipocyte formation was confirmed using Oil Red O (Merck) staining, while osteoblast differentiation was revealed with Alizarin Red S (Merck) staining.

### 2.3. Three-Dimensional Aggregates

#### 2.3.1. Hepatic Spheroid Formation

The cells were trypsinized, resuspended in HMM complete medium, and then seeded (2 × 10^6^ cells/well) in Aggrewell^TM^ microplates (STEMCELL Technologies), composed of approximately 1200 microwells for each of the 24 wells. Before seeding, the plates were treated with Aggrewell^TM^ Rinsing Solution (STEMCELL Technologies) to inhibit cell attachment. After seeding the cells, the plates were centrifuged at 100× *g* for 6 min to distribute the cells homogeneously in the microwells. The plates were then incubated at 37 °C. Half of the medium was changed every other day. Three-dimensional organization started within few hours from seeding, and spheroids formed within 48 h.

#### 2.3.2. Liver Organoid Formation

To generate liver organoids, hiHeps at dd9 of differentiation were mixed with HUVECs and hiMSCs according to the ratio hiHeps:HUVECs:hiMSCs = 10:7:2 [8] and resuspended in 50% HMM, 50% EGM − 2 + 5% FBS. Cell suspension (2 × 10^6^ cells/well) was subsequently seeded in every AggreWell^TM^ microplate, centrifuged at 100× *g* for 6 min, and incubated at 37 °C. Half of the medium was changed every other day. Three-dimensional organization started within few hours, and organoids formed within 48 h.

### 2.4. Dynamic Culture

Dynamic culture of the organoids was conducted in a LiveBox bioreactor (LB; IVTec, Massarosa (LU), Italy). The bioreactor, composed of a transparent chamber, features flow inlet and outlet for perfusion with culture media. The system reproduces the typical volume of a 24-well plate. All the components were autoclaved before use. The content of aggregates from one Aggrewell^TM^ microplate was put in the LB bioreactor 48 h after seeding. In order to avoid loss of aggregates due to dynamic flow, they were resuspended in 1.5% alginate dissolved in 20 mM HEPES-NaOH at pH 7.4, filtered through a 0.45 μm filter, and autoclaved. The aggregates were collected from the Aggrewell^TM^ microplate, and the pellet was resuspended in 1 mL of alginate. To obtain the beads, the cell suspension was placed in a syringe provided with a 21G needle (Terumo, Tokyo, Japan), dropped in a solution of CaCl_2_ at 102 mM under magnetic stirring, and then left for 10 min at 4 °C. The aggregates were then cultured for 96 h at a flow rate of 330 mL/min, and half of the medium was changed 48 h after the starting point (Figure 1).

### 2.5. Flow Cytometry

Cells were detached and resuspended in phosphate-buffered saline (PBS) containing Ca^2+^/Mg^2+^ (PBS+) and 2% bovine serum albumin (BSA). The cell suspension was incubated with the indicated specific antibodies (all from Biolegend, San Diego, CA, USA) for 1 h at 4 °C. Cells were pelleted, washed with PBS/BSA, and then analyzed using FACSCanto^TM^ (BD Bioscience, San Jose, CA, USA).

### 2.6. Viability Assay

The viability assay was performed using CellTiter-Glo^®^ 3D Cell Viability Assay following the manufacturer’s instructions (Promega). Thirty aggregates for each condition were collected in duplicate and measured. ATP-derived luminescence was evaluated and expressed in relative units (RLU). Cell viability was compared to an ATP standard curve.

### 2.7. RNA Extraction and RT-PCR Analysis

Total cellular RNA was extracted using a Quick-RNA^TM^ MiniPrep Kit (Zymo Research, Irvine, CA, USA). Three-dimensional structures were conserved in TRIzol Reagent (Thermo Fisher Scientific) and homogenized with 1 mL Tissue Grinder Potter-Elvehjem (BioSigma, Cona, Italy) for 15 min on ice. The sample was then passed in a 1 mL insulin syringe (Terumo, Tokyo, Japan) 10 times, moved to a new tube, and centrifuged for 10 min at 12,000× *g* at 4 °C. The supernatant was collected, and chloroform was added. Then, the sample was inverted 10 times, left for 4 min at RT, and centrifuged for 20 min at 12,000× *g* and 4 °C. The phase containing RNA was precipitated using isopropanol and centrifuged for 20 min at 12,000× *g* and 4 °C. The pellet was washed and resuspended in 30 μL of RNAse-free water. The quantity and purity of RNA were measured using a NanoDrop spectrophotometer (Thermo Fisher Scientific). Reverse-transcriptase polymerase chain reaction (PCR) of 1 μg of total RNA was carried out using an iScript^TM^ cDNA Synthesis Kit (Bio-Rad, Hercules, CA, USA). For the PCR assay, cDNA was mixed with DreamTaq^TM^ PCR Master Mix (Thermo Scientific), specific primers, and RNase-free water (Bio-Rad).

PCR was performed on Thermocycler for PCR (Bio-Rad) for 5 min at 95 °C followed by 35 cycles (95 °C for 45 s; 60 °C for 45 s; 72 °C for 45 s) and a last cycle at 72 °C for 2 min. PCR products were visualized with agarose/Gel Red (Sigma, Saint Louis, MO, USA) electrophoresis.

For the quantitative PCR (qPCR) assay, cDNA was mixed with SYBR^®^ Select Master Mix (Thermo Fisher Scientific), specific primers, and RNase-free water (BIO-RAD). qPCR was performed on an Applied Biosystem ViiA^TM^ 7 Real-Time PCR System (Thermo Fisher Scientific) for 10 min at 95 °C followed by 40 cycles (95 °C for 15 s; 60 °C for 1 min) and a melt curve stage at 95 °C for 15 s; All PCR reactions were performed in triplicate. The mean cycle threshold (Ct) values were normalized to endogenous control β-actin and analyzed using the comparative ΔCt method. The following primers were used to assess the expression of specific genes: octamer-binding transcription factor 3/4 (Oct 3/4) (For: 5′ GGGTTTTTGGGATTAAGTTCTTCA, Rev: 5′ GCCCCCACCCTTTGTGTT); Nanog (For: 5′ AGGAAGACAAGGTCCCGGTCAA, Rev: 5′ TCTGGAACCAGGTCTTCACCTGT); hepatocyte nuclear factor 4 (HNF4) (For: 5′ TGCGACTCTCCAAAACCCTC, Rev: 5′ TGATGGGGACGTGTCATTGC); alpha-fetoprotein (AFP) (For: 5′ AAGTTTAGCTGACCTGGCTACC, Rev: 5′ TGCAGCAGTCTGAATGTCCG); albumin (For: 5′ TCTTCTGTCAACCCCACACG, Rev: 5′ GCAACCTCACTCTTGTGTGC); serpin A1 (For: 5′ TCCGATAACTGGGGTGACCT, Rev: 5′ AGACGGCATTGTCGATTCACT); actin beta (ACTB) (For: 5′ CACTCTTCCAGCCTTCCTTC, Rev: 5′ AGTGATCTCCTTCTGCATCCT).

### 2.8. Secreted Protein Quantification

Cell conditioned medium was collected every other day starting from dd5 of iDE and subsequently from all culture conditions. The presence of alpha-fetoprotein (AFP) was quantified using a Cobas analyzer (Roche, Basel, Switzerland). Human alpha-1 antitrypsin (AAT) was quantified with a sandwich ELISA. Further, 96-well half-volume high-binding plates (Corning) were coated overnight with 2 μg/mL polyclonal anti-mouse IgG (Sigma), saturated for 1 h at 37 °C with blocking buffer (PBS, 0.25% BSA, 0.05% Tween-20) followed by incubation with conditioned medium of anti-human α1 antitrypsin (AAT) monoclonal antibody 3C11 hybridoma cells, which were kindly provided by Prof. David Lomas (University College London, UK) [13,14]. After washing with PBS/0.05% Tween-20 (PBS-T), serial dilutions of the purified plasma AAT standard and of the cell culture media were added to the plates and incubated at 37 °C for 1 h. After washing with PBS-T, the wells were incubated for 1 h at 37 °C with sheep anti-AAT-HRP-conjugated antibody (Abcam, Cambridge, UK) diluted in blocking buffer, further washed, and revealed with 3,3′,5,5′-tetramethylbenzidine (TMB) substrate (Merck). The reaction was blocked by adding 3 M HCl, and absorbance at 450 nm was measured using an EnSightTM plate reader (PerkinElmer, Waltham, MA, USA).

### 2.9. Immunohistochemistry

For immunohistochemistry analysis, liver organoids were collected and centrifuged at 3000× *g* × 10 min, and a compacting solution consisting of 75% methanol (Carlo Erba, Cornaredo, Italy), 20% chloroform (Sigma) and 5% acetic acid (Carlo Erba) was added to the pellet for 20–30 min as already described [15]. The compacting solution was subsequently discarded, the pellet was moved to a histology cassette, stained with hematoxylin and eosin, covered with 2.5% agarose, and preserved in formalin until it was processed for paraffin inclusion. Endogenous peroxidase activity was blocked with 0.3% H_2_O_2_ (Sigma Aldrich) in methanol for 20 min. Antigen retrieval was performed using a microwave oven in 1 mM EDTA at pH 8.0 (Carlo Erba) or in 1 mM citrate buffer at pH 6 (Carlo Erba). Two-micron sections were cut, washed in TBS at pH 7.4 (Carlo Erba), and incubated for 1 h with the following primary antibodies diluted in TBS/1% BSA (Merck): anti-human albumin (ThermoFisher Scientific), anti-CD90 (Abcam rabbit monoclonal), and anti-CD31 (Novocastra) antibodies. The signal was revealed using DAKO Envision + System-HRP Labelled Polymer Anti-Mouse or Anti-Rabbit, followed by diaminobenzydine (DAB) as chromogen and hematoxylin as counterstain. Images were acquired with a Nikon DS-Ri2 camera (4908 × 3264, full pixel) mounted on a Nikon Eclipse 50i microscope equipped with Nikon Plan lenses (×40/0.65) using NIS-Elements 4.3 imaging software (Nikon Corporation, Tokyo, Japan).

### 2.10. Immunofluorescence Staining

To perform immunofluorescence experiments, cells were fixed using Immunofix (Bio-Optica) for 15 min, washed, and maintained in PBS+ at 4 °C. Before antibody staining, cells were permeabilized with PBS+ containing 0.5% Triton^TM^ X-100 (Sigma) for 10 min, then incubated with a blocking solution (PBS+ containing 1% BSA (Sigma) and 2% Donkey Serum (Merck)) for 1 h at RT, and washed with PBS+ containing 0.2% Tween 20 (Merck). At this point, samples were incubated overnight at 4 °C with the following primary antibodies diluted in blocking solution: anti-forkhead box A2 (FOXA2; rabbit polyclonal IgG; 1:400; Thermo Fisher Scientific (720061)), anti-SRY-box transcription factor 17 (SOX17; mouse monoclonal IgG2b (OTI2G8); 1:400; Thermo Fisher Scientific (MA5-24891)), or anti-HNF4 (rabbit monoclonal IgG (F.674.9); 1:400; Thermo Fisher Scientific (MA5-14891)). After extensive washing, Alexa fluorophore-conjugated secondary antibodies (donkey anti-rabbit IgG 594; 1:600; Thermo Fisher Scientific (A-21207); donkey anti-mouse IgG 488; 1:600; Thermo Fisher Scientific (A-21202)) were added and incubated for 1 h at RT. Nuclei were counterstained with 4′,6-diamidino-2-phenylindole (DAPI) (Thermo Fisher Scientific) for 10 min at RT. Images was acquired at ×20 and ×63 magnification with a Zeiss Fluorescence Axiovert 200 M microscope (Carl Zeiss, Oberkochen, Germany).

### 2.11. Statistical Analysis

All data were expressed and plotted as means ± standard errors of the means (SEMs). All statistical analyses were performed with OriginLab. Comparison within different conditions was calculated using one-way ANOVA, and statistical significance was defined at * *p*-value < 0.05.

## 3. Results

### 3.1. Differentiation of Mesenchymal Stem Cells from Induced Pluripotent Stem Cells

We previously isolated human MSCs from bone marrow and from different adipose tissues, concluding that MSCs present distinct characteristics depending on their isolation source [16]. Since our aim was to include MSCs in liver organoids, we decided to differentiate them starting from hiPSCs. As illustrated by the flow cytometry analysis in Figure 2A, hiMSCs showed a remarkable expression of the mesenchymal specific markers CD105, CD90, and CD73, while they were negative for the expression of CD34, specific to the hematopoietic lineage. Moreover, hiMSCs showed osteogenic and adipogenic differentiation potential, as indicated by the calcium deposits highlighted with Alizarin Red S staining (Figure 2C) and by the neutral triglyceride and lipid deposits present in adipocytes detected with Oil Red O staining (Figure 2D), respectively. These specific markers’ expression, together with cell morphology (Figure 2B) and adherence to plastic, as well as their proliferation for more than 10 passages, categorized these cells as mesenchymal stem cells.

### 3.2. Differentiation of Hepatic Cells from Induced Pluripotent Stem Cells

To generate hepatic cells from hiPSCs, we followed the protocol illustrated in Figure 3A, which comprises preliminary steps lasting 5 days, leading to the differentiation toward hiDE, followed by the specification of the hepatic lineage. The characterization of hiDE cells is illustrated in Figure 3. In parallel to an evident change in cell morphology (Figure 3D), there was a progressive down-regulation of pluripotency genes, represented by OCT4 (Figure 3B), paralleled by an upregulation of specific markers, such as HNF4 (Figure 3C), involved in the transition from endodermal cells to hepatic progenitors. Furthermore, hiDE cells were stained positive for the endodermal surface markers C-X-C Motif Chemokine Receptor (CXCR) 4 and Cluster of Differentiation (CD) 117 (Figure 3E) in flow cytometry and for the endodermal markers FOXA2 and SOX17, both involved in the development of the hepatic-cell lineage, in immunofluorescence (Figure 3F). We continued the differentiation up to dd23, and on dd21, we performed several analyses in order to determine if hepatocyte specification had occurred. We evaluated the presence of mRNA encoding two of the most abundant proteins produced by hepatocytes: AFP, the fetal albumin counterpart, and α1 antitrypsin (AAT), a serine proteinase inhibitor. The results shown in Figure 3G demonstrate an increase in the mRNA expression of both genes starting from dd9 and more pronounced on dd21, when the expression levels were slightly higher than the levels observed in hepatoma HepG2 cells. Moreover, the encoded proteins were correctly secreted into the culture medium (Figure 3H). AFP started to be present in the medium around dd11, soon after hepatic specification, and decreased after dd15. Also, AAT appeared on dd11 but showed a progressive increase over time. These trends suggest the presence of liver cells, which undergo a maturation process that involves the switch between AFP and albumin production.

### 3.3. Formation of 3D Aggregates

After successfully deriving hiMSCs and hiHeps from hiPSCs, we combined these two cell types with endothelial cells (HUVECs) in a predetermined ratio [8], and we cultivated the mixed cells in Aggrewell^TM^ culture microplates to obtain 3D structures. The mixed culture (Figure 4A) demonstrated the capacity of cells to self-aggregate and to form three-dimensional structures within 48 h of co-culture (Figure 4B). Remarkably, these structures maintained their three-dimensionality when removed from the Aggrewell^TM^ microplates (Figure 4C,D). To analyze the composition of the aggregates 48 h after seeding, we conducted immunohistochemistry experiments. Hematoxylin–eosin staining (Figure 4E) revealed the cellularity of the three-dimensional structures. Additionally, in order to examine the organization of the three cell types within the organoids, we utilized anti-AFP antibody (blue) to identify hepatocytes, anti-CD31 antibody (red) for endothelial cells, and anti-CD90 antibody (brown) for mesenchymal cells (Figure 4F). These immunostaining procedures confirmed the presence of all three cell types within the architectural framework of the organoids. Notably, 48 h post-formation, the cellular arrangement appeared to be stochastic, with mesenchymal cells being predominantly located at the core, while hepatocytes and endothelial cells were more prevalent in the outer regions of the structure.

### 3.4. Three-Dimensional Aggregates in Dynamic Culture

After confirming the self-aggregation capacity of hiHeps, hiMSCs, and HUVECs to form heterotypic 3D aggregates, we also obtained spheroids composed of only hiHeps that could serve as controls for the following experiment. These homotypic 3D structures were also formed using AggreWell^TM^ microplates. The size of the organoids or the spheroids was about 200 μm. These dimensions are highly deleterious to their long-term culture as required in pharmacological or toxicological testing, as the static culture condition poorly permits the diffusion of nutrients into the center of the 3D structures. To overcome this hurdle, we put the aggregates in the dynamic framework of a LiveBox millifluidic bioreactor (IVTech), where the aggregates could be subjected to a continuous flow of culture medium. Moreover, these dynamic conditions better reproduced the sinusoidal blood flow impact on hepatocyte functions in the organism. To avoid the loss of samples due to medium flow, we entrapped the aggregates into 1.5% alginate beads (Figure 5).

To analyze the efficiency and effectiveness of the dynamic culture conditions, we compared cell viability under three different culture conditions: we left some aggregates in an Aggrewell^TM^ microplate as the control, unperturbed conditions (CTRL1); for the second condition, 48 h after aggregation, 3D structures were collected and entrapped in alginate beads and cultured under static conditions (CTRL2); finally, the aggregates were placed in the culture chamber of a LiveBox bioreactor (LB) and cultured under dynamic conditions. Samples were analyzed after 96 h (4 days).

We first analyzed cell viability using an ATP luminescent kit expressly designed for microtissues. To set up a standard curve, we measured the luminescence of 25,000 and 50,000 live trypan blue-negative cells, either hiHeps alone or mixed with HUVECs and hiMSCs in the same ratio as in organoids. Then, to reduce the sample’s variability, we quantified luciferase in 30 aggregates, in duplicate, for each experimental condition. We measured ATP-derived luminescence (RLU) at the moment of aggregate collection, t_0_ (48 h after seeding), and after 96 h of culture under the three different conditions (Figure 6A).

As shown, aggregates’ viability was reduced under all conditions. However, while dynamic culture strongly reduces spheroid survival, the same condition represents the best setup for organoid culture.

Organoids are formed following the aggregation of three cell types that could behave differently in dynamic culture. Therefore, we must ensure that the liver cells can survive under these conditions. To better assess the characteristics of our 3D aggregates, we quantified the mRNA expression levels of hepatic genes in the collected aggregates after one week of dynamic culture. All the expression levels of the hepatic genes showed a strong increase in the final sample, confirming the survival of the hepatic lineage (Figure 6B). Moreover, the large increase in the expression levels of *ALB* mRNA over time is particularly promising, as it strongly suggests hepatic-cell maturation.

Based on these promising results, we aimed to validate the data by collecting the organoids after one week of culture in the bioreactor. The collected organoids were embedded in paraffin, sectioned, and subjected to analysis using hematoxylin–eosin (H&E) staining. Additionally, immunohistochemical staining was performed using antibodies specific to the three cell types. H&E staining (Figure 6C) showed the presence of viable cells throughout the center of the aggregates. Of note, minimal presence of apoptosis/necrosis could be seen. Additionally, the organoids demonstrated ALB production, as depicted in Figure 6D. To investigate the arrangement of the three cell types within the organoids, histological sections were stained using three specific antibodies, anti-CD31 for HUVECs (stained in red), anti-CD90 for hiMSCs (stained in brown), and AFP for hiHeps (stained in blue), as shown in Figure 6E. The presence of the three colors indicates the survival of all cell types under dynamic conditions. The images suggest a structural organization of the cell types, where hiHep cells were predominantly located on the surface, while hiMSCs were neatly arranged to form a structural support. Moreover, HUVECs seemed to have organized themselves in circular structures, probably the beginning of capillary neoformation.

## 4. Discussion

The process of bringing new drugs to the market starts with the discovery of new molecules, followed by several steps of safety and efficacy trials prior to their approval by national and international regulatory agencies, such as the U.S. Food and Drugs Administration. Typically, this procedure takes several years, with a significant financial investment [17,18].

In drug research, the major issue is the risk associated with hepatotoxicity, which is the main reason for the withdrawal of drugs from the market [17,18,19].

The animal model is considered the gold standard in drug preclinical testing, but it presents several issues, because animals frequently have a different hepatic metabolism compared with humans. Consequently, the transposition of results from animal studies to human therapy is difficult. These differences represent an important risk factor in preclinical trials, as the predictive level of hepatotoxicity obtained from animal tests is only around 50% [20]. Furthermore, the extended lead times and relatively high costs associated with animal models require the development of new, faster, and cheaper in vitro models capable of fulfilling the 3R principles of replacement, reduction, and refinement of animal models [3].

The discovery of hiPSCs has opened new possibilities for their application as a model in drug testing. Currently, well-established protocols are described for quite easily deriving human hepatocytes from hiPSCs [21,22]. iPSC-induced hepatocytes could represent an excellent cell model for drug testing and screening, particularly because they can be derived from patients autologous cells [23]. Nevertheless, it is important to note that the liver is a highly complex organ with specific architecture, organization, and functions that are correlated to the several cell types that populate it. Therefore, the self-assembling 3D structure of organoids, composed of different cell types and representing the main components of the organ, could be considered as an initial step toward mimicking the complex structure of the liver and its physiologically relevant impact on hepatic functions [6,24].

In our study, we focused on deriving and characterizing both hiMSCs as the structural component and hiHeps as the functional part of the organoids. We also incorporated HUVECs as the endothelial component. We know that endothelial cells can also be derived from hiPSCs, but a robust protocol for differentiating them into liver sinusoidal endothelial cells has not been published yet.

Firstly, we successfully demonstrated the derivation of both hiMSCs and hiHeps. To confirm the differentiation of hiMSCs, we observed the loss of expression of pluripotency genes, the acquisition of a new cell morphology, their ability to grow on plastic surfaces, the expression of specific markers, and the potency to differentiate into both osteoblasts and adipocytes. Regarding hiHep differentiation, we achieved the desired results, as the morphological and biochemical changes recapitulated the ones that occur during hepatogenesis [25]. hiPSCs subjected to hepatic differentiation exhibited an increased cytoplasm–nucleus ratio and assumed the peculiar polygonal morphology. Furthermore, we observed the presence, from dd5, of major key regulators of liver development, such as FOXA2, SOX17, and HNF4, which are completely absent in undifferentiated hiPSCs. Indeed, the decrease in pluripotency genes was paralleled by the increase in HNF4a, which is the master regulator of hepatoblastic differentiation, as well as the appearance of the DE-specific surface markers CXCR4 and CD117. The resulting hepatocyte-like cells, which emerged around dd11, just when conversion from liver progenitors to liver cells was expected, produced both AFP and AAT, which were secreted into the culture medium.

The initial increase in AFP concentration, followed by its successive decrease in the culture medium, could be an indication of the maturation of liver-like cells. Under physiological conditions, AFP is progressively deactivated and replaced by ALB in adult hepatocytes. Similarly, the production of AAT, with a progressive increase along the entire differentiation process, recalls the behavior observed during the physiological maturation of hepatocytes. All acquired data supported the success of the differentiation protocol, but they also show the limitation of 2D hiHep culture. While 2D culture is essential to obtaining liver-like cells from hiPSCs, it is not sufficient to reach a stable and mature phenotype.

To overcome the limitations of 2D cultures and promote a more physiologically relevant model, we mixed the three cell types (hiHeps, hiMSCs, and HUVECs) to create a self-assembled three-dimensional organoid with a specific architecture that mimicked the liver structure. While other laboratories build their hepatic organoids with the aim of transplanting them into an organism, our goal is to use the organoids in vitro for cell culture experiments. Our aim is to replace the use of animals to define hepatic pharmacology and toxicity with an alternative approach.

For this purpose, we miniaturized the aggregates, seeding the cells in Aggrewell^TM^ microplates, following the previously described ratio that closely recapitulates the liver composition [8]. Initially, we created spheroids using hiHep cells only, but data demonstrate that monotypic culture shows feasibility limits. For this reason, we assembled organoids, adding mesenchymal cells, also derived from hiPSCs, and endothelial cells. Furthermore, since the size of these 3D structures could hamper the perfusion of nutrients, especially in long-term culture, we introduced the heterotypic aggregates in a bioreactor system. The bioreactor facilitated the continuous recirculation of the culture medium, increasing mass transfer and ensuring sufficient nutrient supply to the center of the 3D structures. To prevent the aggregates from being carried away by the flow of the medium, we encapsulated them in alginate beads. Alginate is a compound that, by gelling, forms a stable scaffold without causing any damage to the cellular structures.

Under these culture conditions, we observed higher viability of the organoids compared with spheroids. With histological analysis and subsequent immunolocalization, we demonstrated the survival, aggregation, and structural organization of all cell types. In several organoids, we observed the presence of a fibrotic core or a ring-like structure, which was probably due to hiMSC matrix production. This fibrotic structure appeared to provide support for the hiHeps arranged in the outer area of the organoids. The endothelial cells, on the other hand, appeared to be dispersed throughout the organoids, forming small clusters, indicating the initiation of organization into sinusoidal-like structures. Notably, under these conditions, ALB expression appeared to be elevated, suggesting that the dynamic culture could enhance and support the maturation of the hepatocytes involved.

Overall, this approach allows for the maintenance and proper functioning of the organoids, providing an improved environment for their growth and development. The combination of miniaturized aggregates, the incorporation of multiple cell types, and the use of a bioreactor system with encapsulation in alginate beads represent an innovative strategy to overcome challenges associated with nutrient perfusion and the maintenance of the structural integrity of organoids in long-term culture.

## 5. Conclusions

Based on the tissue engineering concept, we developed a complex organotypic model, derived from hiPSCs, in both static and dynamic systems to discover its application benefits in research. We can conclude that the 3D cell culture of heterotypic cells maintained in a dynamic environment offers a much more realistic hepatic model than the classical two-dimensional culture, thus offering us the opportunity to learn more about the physiological and pathological phenomena on which we can intervene pharmacologically. In this manuscript, we focus on more differentiated hiHeps, but this hepatic model can also be used to study hepatogenesis, since we succeeded in reproducing hiHeps from the less differentiated state of hiPSC to endoderm specific cells, hepatoblasts, and hepatocytes.

## Figures and Tables

**Figure 1 biomedicines-11-02114-f001:**
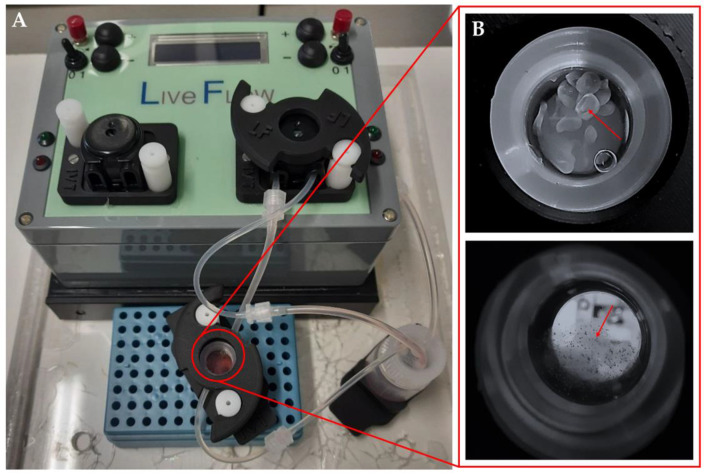
Apparatus for biodynamic culture. The bioreactor by IVTech LiveBox (LB) (**A**). Magnification of the culture chamber with alginate beads (red arrow, top) or alginate-containing organoids (red arrow, bottom) (**B**).

**Figure 2 biomedicines-11-02114-f002:**
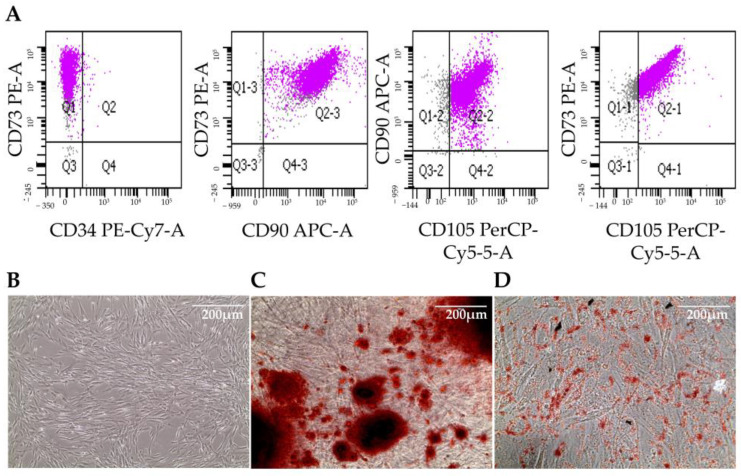
Characterization of hiMSCs. Flow cytometry scatters of hiMSCs using the indicated antibodies (**A**). Cell morphology (**B**). Alizarin Red staining (**C**). Oil Red O staining (**D**). Scale bars = 200 μm.

**Figure 3 biomedicines-11-02114-f003:**
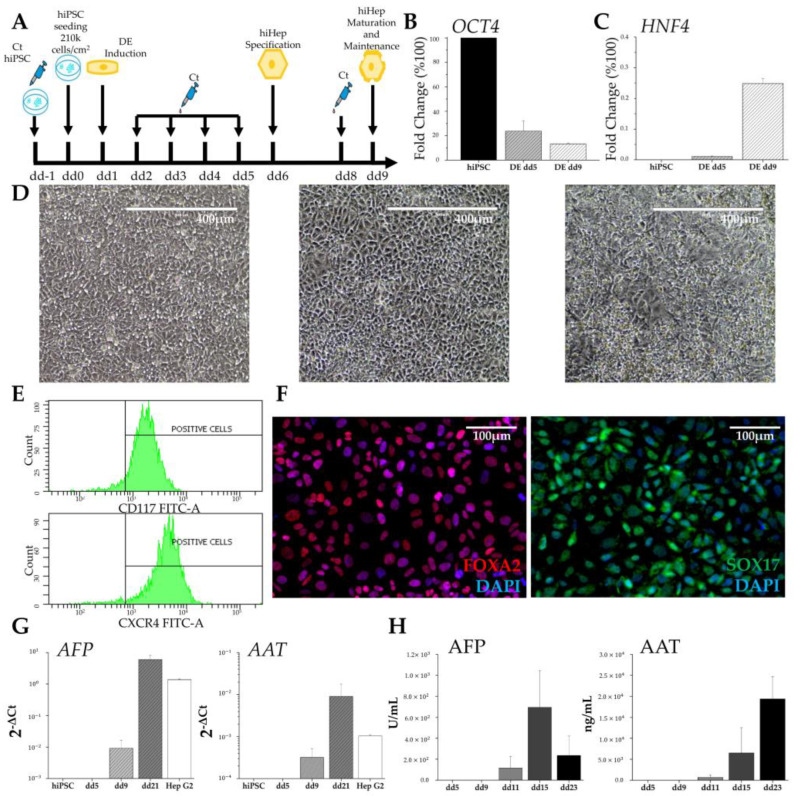
Characterization of human-induced hepatic-like cells. Protocol used for differentiating hiPSCs into hepatic-like cells (**A**). RT-PCR analysis of pluripotency gene OCT4 (**B**) and of the hepatic lineage marker HNF4 (**C**); results are expressed as the fold-change percentage of definitive endoderm (DE) on dd5 and dd9 with respect to hiPSCs values ± SEMs, normalized to β-actin of 2 independent differentiations (*n* = 2). Bright-field images of progressive changes in cell morphology during hepatic endodermal induction: hiPSCs on dd0, followed by DE on dd5 and on dd9, respectively; scale bars = 400 μm (**D**). Flow cytometry for surface markers CD117 and CXCR4 on dd9 (**E**). Immunocytochemistry on dd9 showing the endodermal markers FOXA2 (red) and SOX17 (green); nuclei were counterstained with DAPI (blue); scale bars = 100 μm (**F**). RT-PCR analysis of hepatic genes AFP and AAT evaluated in hiPSCs on dd5, dd9, and dd21 using Hep G2 cells as control; results are expressed as 2^−ΔCt^ ± SEMs, normalized to the β-actin value from at least two independent differentiations (*n* ≥ 2) (**G**). Hepatic protein secretion measured in culture medium collected on dd5, dd9, dd11, dd15, and dd23; AFP and AAT were quantified from at least two independent differentiations (*n* ≥ 2) (**H**).

**Figure 4 biomedicines-11-02114-f004:**
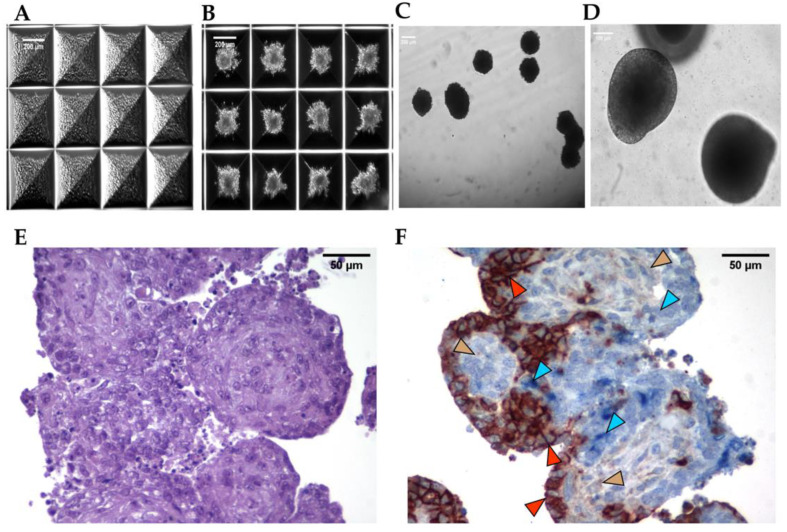
Aggregate formation. Representative bright-field microscopic images overviewing aggregate formation in Aggrewell^TM^ microplates starting from mixed hiHep, hiMSC, and HUVEC single-cell seeding (**A**); after 48 h (**B**); scale bars = 200 μm. Bright-field microscopic images of floating aggregates after 7 days of culture at lower (scale bar = 200 μm) (**C**) or higher (scale bar = 100 μm) magnification (**D**). Hematoxylin–eosin staining of aggregates after 48 h of culture; scale bar = 50 μm (**E**). Co-immunostaining using anti-AFP antibody (blue) to label hiHeps, anti-CD-31 antibody (red) to label endothelial cells, and anti-CD-90 antibody (brown) to label hiMSCs. Arrowheads highlight the signals for the investigated molecules: blue indicates hiHep cells; red indicates HUVECs; and brown indicates hiMSCs; scale bars = 50 μm (**F**).

**Figure 5 biomedicines-11-02114-f005:**
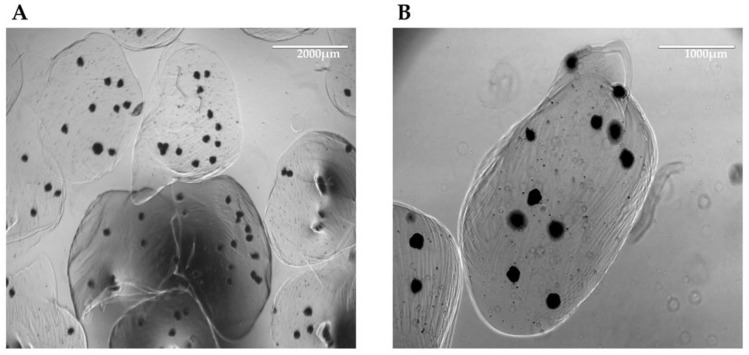
Organoids in alginate beads. Representative bright-field microscopic images of organoids entrapped in 1.5% alginate beads at lower (scale bar of 2000 μm) (**A**) or higher (scale bar of 1000 μm) magnification (**B**).

**Figure 6 biomedicines-11-02114-f006:**
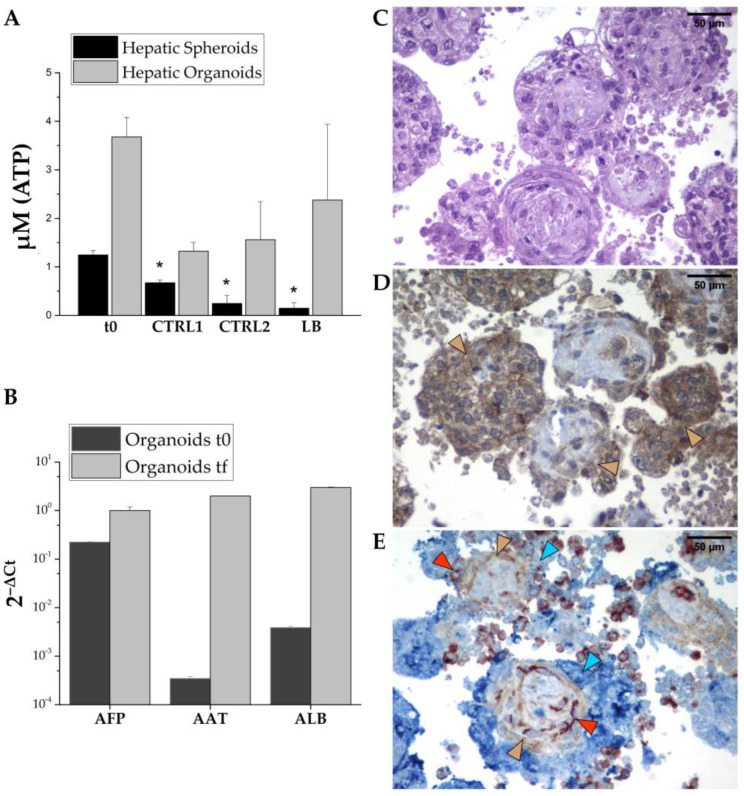
Organoid characterization. Viability assay showing the amount of cellular ATP (μM) in the newly formed hepatic organoids (light-gray bars) or spheroids (black bars); t0 correspond to aggregates 48 h after seeding; CTRL1 corresponds to aggregates in Aggrewell^TM^ microplates without any perturbation; CTRL2 corresponds to half Aggrewell^TM^ aggregates entrapped in alginate beads 48 h after seeding and maintained under static culture conditions; and LB corresponds to half Aggrewell^TM^ aggregates entrapped in alginate beads 48 h after seeding and maintained under dynamic culture conditions. CTRL1, CTRL2, and LB were collected after 1 week of culture. Box plots show the ATP concentration means ± SEM of at least ≥ 2 independent differentiations. Comparison within different conditions was calculated using one-way ANOVA. * *p*-value < 0.05 (**A**). RT-PCR analysis of the hepatic markers *AFP*, *ALB*, and *AAT* on organoids 48 h after seeding (t_0_), and after 1 week of culture in Aggrewell^TM^ (t_f_); results are expressed as the means of technical triplicate 2^−ΔCt^ ± SEM normalized to β-actin of a pool of ~1200 organoids (**B**). Hematoxylin–eosin staining of heterotypic organoids composed by hiHeps, hiMSCs, and HUVECs after 96 h of culture under dynamic conditions; scale bar = 50 μm (**C**). Albumin immunolocalization using anti-albumin antibody; brown arrowheads indicate albumin-positive cells; scale bar = 50 μm (**D**). Co-immunostaining using anti-AFP antibody (blue) to label hiHeps; anti-CD-31 antibody (red) to label endothelial cells and anti-CD-90 antibody (brown) to label hiMSCs. Arrowheads highlight the signals for the investigated molecules: blue indicates hiHep cells; red indicates HUVECs; and brown indicates hiMSC; scale bar = 50 μm (**E**).

## Data Availability

Not applicable.

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
