# Peer review of "Human iPSC-Derived 3D Hepatic Organoids in a Miniaturized Dynamic Culture System"

_biomedicines, 2023, doi:10.3390/biomedicines11082114_

Round 1
Reviewer 1 Report
Very interesting work presented. The following comments and questions arose:
1. It is necessary to submit a section of statistical data analysis. Indicate what indicators were used to build graphs, how many samples were taken in each analysis.
2. I would like to see how the used cells were distributed in the resulting organoids.
Author Response
First, we would like to thank you for your valuable time and effort used to review the manuscript. We sincerely appreciate the precious comments and suggestions which helped us in strongly improving the quality of the manuscript.
Here is our reply to your requests:
- It is necessary to submit a section of statistical data analysis. Indicate what indicators were used to build graphs, how many samples were taken in each analysis.
We add a section in Material and Methods where we describe the statistical data analysis. The number of samples can be found in the text or in the figures’ caption.
- I would like to see how the used cells were distributed in the resulting organoids.
In Figure 4 and 6 we add colored arrowhead that, identifying the different cell type, show the relative distribution.
Reviewer 2 Report
In general, this article has unclear images with many details that are difficult to identify. The methods used to culture the organoids are almost identical to those used in other papers, and the identification of the organoids is almost completely omitted, raising doubts as to whether the team has successfully cultured liver organoids.
Several points should be noted:
1. The cultivation methods for hiMSC derived from hiPSC in method 2.2.2 are not specific enough.
2. The captions under Figure 3 provide insufficient information. For example, it is unclear how many data sets were used to generate the bar graphs in Figures 3C, 3G and 3H and what the corresponding p-values are. In Figure 3D it is unclear which cell types are represented in the images for dd0, dd5 and dd9. Regarding the gene expression analysis in Figures 3E and 3F, it is not mentioned on which day the samples were collected. These details need to be clarified.
3. The magnification of the low-power and high-power images in Figures 4 and 5 should be clearly stated. The microscopic images in Figure 4 show only aggregate spheres formed after single cell seeding, but do not show the formation of organoids.
4. In the Results section, while 3.1 mentions the identification of mesenchymal stem cells derived from hiPSC and 3.2 describes the formation and identification of hiHeps, I believe that 3.3 should have discussed the formation process and identification of liver organoids after mixing the three cell types (hiMSC, HUVEC, hiHeps). However, this article jumps directly to the section on dynamic cultivation, completely omitting the organoid cultivation part, which casts doubt on whether the author's team has successfully cultured liver organoids.
5. The same problem applies to Figure 6. The caption does not provide enough information. For example, what does "Ctrl1" in Figure 6A represent in the experimental setup? The article also does not mention what "CTRL2" and "LB" refer to. Necessary explanations should be given in the figure caption. It is also important to state the number of data sets used to generate the bar graphs and the corresponding p-values. All these details should be clearly stated.
6. Lines 330-336 belong to the experimental methods and should not be included in the results section.
Author Response
First, we would like to thank you for your valuable time and effort used to review the manuscript. We sincerely appreciate all the precious comments and suggestions which helped us in strongly improving the quality of the manuscript.
Here is the reply to your requests:
- The cultivation methods for hiMSC derived from hiPSC in method 2.2.2 are not specific enough.
We have expanded the cultivation method for hiMSC in method 2.2.2
- The captions under Figure 3 provide insufficient information. For example, it is unclear how many data sets were used to generate the bar graphs in Figures 3C, 3G and 3H and what the corresponding p-values are. In Figure 3D it is unclear which cell types are represented in the images for dd0, dd5 and dd9. Regarding the gene expression analysis in Figures 3E and 3F, it is not mentioned on which day the samples were collected. These details need to be clarified.
We completely change the caption of Figure 3, following in detail reviewer’s suggestions.
- The magnification of the low-power and high-power images in Figures 4 and 5 should be clearly stated. The microscopic images in Figure 4 show only aggregate spheres formed after single cell seeding, but do not show the formation of organoids.
We clearly stated the magnification in Figures 4 and 5 by adding scale bars. Moreover, in Figure 4 we add H&E staining as well as immunolocalization of the different cell types in the formed organoids.
- In the Results section, while 3.1 mentions the identification of mesenchymal stem cells derived from hiPSC and 3.2 describes the formation and identification of hiHeps, I believe that 3.3 should have discussed the formation process and identification of liver organoids after mixing the three cell types (hiMSC, HUVEC, hiHeps). However, this article jumps directly to the section on dynamic cultivation, completely omitting the organoid cultivation part, which casts doubt on whether the author's team has successfully cultured liver organoids.
We add the section 3.3 where we describe the formation of aggregates.
- The same problem applies to Figure 6. The caption does not provide enough information. For example, what does "Ctrl1" in Figure 6A represent in the experimental setup? The article also does not mention what "CTRL2" and "LB" refer to. Necessary explanations should be given in the figure caption. It is also important to state the number of data sets used to generate the bar graphs and the corresponding p-values. All these details should be clearly stated.
We completely changed the text and caption of Figure 6 so that the meaning of CTRL1, CTRL2 and LB was clear.
- Lines 330-336 belong to the experimental methods and should not be included in the results section.
We change the text by deleting what belong to experimental method.
Reviewer 3 Report
Summary: The purpose of the present manuscript has been to develop an in vitro model that can replace the animal models for testing the pharmacological potential and toxicological effects of medicines and other substances on liver cells. This model combines the heterotypic composition of liver cells, namely hepatocytes, endothelial cells and mesenchymal cells, the 3D organization of cells (organoids matrix) as well as microfluidic simulation of sinusoidal flux within the intrahepatic microenvironment.
The authors will find hereinafter my comments and suggestions:
1. The „Abstract” conveys the main findings of the study.
2. „Introduction” section appropriately provide the scientific context of the approached topic and „Results” and „Discussion” sections are significant to the approached field.
3. The authors are advised to expand the abbreviations at their first use within the manuscript. For instance, EGM-2, BSA, PBS-T, TMB, anti-AAT-HRP, FOXA2, SOX17, HNF4, DAPI, etc.
4. The scale bars must be added in figures 2B, 2C, 2D; 3D, 3F; 4A-D; 5A-B; 6C, 6D, 6E.
5. For a better visual impact to the reader please redraw the graphics from figures 6A and 6B.
6. Please, also use arrows, arrowheads in Figures 6D-6E to highlight the positive signals for investigated molecules.
Author Response
First, we would like to thank you for your valuable time and effort used to review the manuscript. We sincerely appreciate all the precious comments and suggestions which helped us in strongly improving the quality of the manuscript.
Here is the reply to your request:
- The authors are advised to expand the abbreviations at their first use within the manuscript. For instance, EGM-2, BSA, PBS-T, TMB, anti-AAT-HRP, FOXA2, SOX17, HNF4, DAPI, etc.
We expand all the abbreviations.
- The scale bars must be added in figures 2B, 2C, 2D; 3D, 3F; 4A-D; 5A-B; 6C, 6D, 6E.
We added scale bars in the captions of the different Figures.
- For a better visual impact to the reader please redraw the graphics from figures 6A and 6B.
We redraw the graphics from Figure 6A and 6B.
- Please, also use arrows, arrowheads in Figures 6D-6E to highlight the positive signals for investigated molecules.
We use colored arrowheads to highlight the positive signals.
Round 2
Reviewer 1 Report
All comments were answered satisfactorily.
Author Response
Thank you
Reviewer 2 Report
In general, the revised manuscript has significantly improved the main problems mentioned in the previous review, and the overall structure is more complete, logical and informative than before. However, I still have the following questions about the revised article:
1. In the introduction, the author mentioned that the purpose of dynamic culture of liver organoids is to perform pharmacological tests, so the function of liver organoids should not only be verified by AFP, ALB and AAT, but also by drug metabolism related indicators such as CYP3A4, CYP2D6, etc.?
2. The immunostaining of hiHeps, hiMSC and HUVEC in Figure 6C, D and E does not indicate the formation of organoids, but only the formation of three types of cell aggregates. If the formation of liver organoids is to be demonstrated, an assay of the types of cells contained in the liver, such as the markers for hepatic sinusoidal endothelial cells and markers for hepatic stellate cells, etc., should be performed and compared with primary hepatocytes.
3. The authors say in the abstract that the dynamic culture can be maintained for 15 days, but in the text, there is no data to support this, and only 1 week at most.
Author Response
In general, the revised manuscript has significantly improved the main problems mentioned in the previous review, and the overall structure is more complete, logical and informative than before. However, I still have the following questions about the revised article:
- In the introduction, the author mentioned that the purpose of dynamic culture of liver organoids is to perform pharmacological tests, so the function of liver organoids should not only be verified by AFP, ALB and AAT, but also by drug metabolism related indicators such as CYP3A4, CYP2D6, etc.?
The aim of this work is to demonstrate the possibility of developing mini liver organoids including hiPSCs truly differentiated into either mesenchymal (hiMSC) or epithelial (hiHep) cell derivatives. Therefore, we monitored the ability of the cells to express both early hepatocyte marker such as AFP and late markers such as ALB and AAT. This demonstration is important in showing how heterotypic three-dimensional structures can contribute to a more mature phenotype, which is a key aspect when using hiPSCs as a model. The switch we observed between AFP and ALB serves as a good indicator of the health of our cells, that we did not observe in spheroids due to the absence of supporting cells. Furthermore, in this work, we aimed to demonstrate the efficacy of a dynamic millifluidic culture system. This system provides the operator with full control of the flow speed, supporting its structure, morphology, and functions. The liver has more than 500 functions and it is true that it is necessary to ascertain the metabolism of drugs and equally important functions. However, since this is a preliminary study, we chose to focus only on hepatocyte protein production, markers commonly used to demonstrate the efficacy of in vitro models. In our dynamic 3D culture condition, we observed increased cell viability, as well as enhanced maturation of hepatocytes. These results suggest that our model could be suitable for an application in pharmaco-toxicological studies, where it is often required to test the hepatotoxicity of the drug or substance. We appreciate your suggestion and rest assured that your request will be considered in future work that will focus on the functionality of our model, possibly in conjunction with drug screening applications.
2. The immunostaining of hiHeps, hiMSC and HUVEC in Figure 6C, D and E does not indicate the formation of organoids, but only the formation of three types of cell aggregates. If the formation of liver organoids is to be demonstrated, an assay of the types of cells contained in the liver, such as the markers for hepatic sinusoidal endothelial cells and markers for hepatic stellate cells, etc., should be performed and compared with primary hepatocytes.
Organoids are generally defined as self-organizing 3D tissue structures derived from stem cells [9;10;24]. In our experimental conditions, we proposed different cell configurations: liver spheroids (culture of hiHep monotypic cells) and liver organoids (culture of heterotypic cells, including iPSC-derived mesenchymal stem cells, iPSC-derived hepatocytes, and endothelial cells). All cell types in our 3D structures were identified by immunostaining (see Fig. 4).
Only a fraction of all cells present in the hepatic tissue are represented in these heterotypic cultures, but certainly the most important cell types are present: hiHeps that reproduces the epithelial part of the liver, which makes up about 70% of the total liver cells, while for the non-parenchymal compartment (NPC) of liver cells, we included endothelial cells mimicking the sinusoidal part of the liver (HUVEC) as well as hiMSCs that mimic the mesenchymal derivation of hepatic stellate cells (Kordes et al., 2013).
Therefore, our hiPSC-derived 3D heterotypic cell cultures should be considered as liver organoids, even if the macrophage lineage (Kupfer cells) was not included.
Due to the challenges associated with obtaining human material, hiPSC derivatives represent an important strategy for developing in vitro models.
Figures 6C, 6D and 6E were obtained from the same paraffin-embedded sample, from which various fields were observed. In Figure 6C, H&E staining shows the cellular composition of three-dimensional structures, which is maintained even after one week of dynamic culture. Figure 6D represents immunostaining with an anti-albumin antibody. Supporting the molecular data, the cells were positive for ALB production, which we consider a late/adult liver marker, indicating the mature level achieved by iPSC-derived hepatocytes within our organoids, even after one week of dynamic culture. In addition to staining for albumin, we performed triple co-staining to identify the three cell types used (Fig. 6E). Not only are these markers kept within the 3D structure, but the structures also display a distinctive organization. This autonomous organization brings support cells (hiMSCs) to the center of the organoid architecture and positions epithelial cells (hiHep) in the outermost part, where they intercalate with endothelial cells.
3. The authors say in the abstract that the dynamic culture can be maintained for 15 days, but in the text, there is no data to support this, and only 1 week at most.
Thanks again for the valuable advice: the differentiation time is 15 days but the dynamic culture is only one week. We changed the text in the abstract.
Round 3
Reviewer 2 Report
Thanks to the authors for the careful revision. The authors have used more conservative and precise wording in the paper. Overall, the latest manuscript is an improvement over the previous manuscript.